# Complete molecular spectrum of β-globin gene mutations via direct sequencing identifies seven novel variants in β-thalassemia major

Torin Abdulaziz Sadoon[1]☉, Hemin Esmael Othman 📵[2]☉*

1 Department of Medical Laboratory Technology, College of Health and Medical Technology-Shekhan, Duhok Polytechnic University, Kurdistan Region, Iraq, 2 Department of Biology, College of Science, University of Duhok, Kurdistan Region, Iraq

☉ These authors contributed equally to this work.
* hemin.othman@uod.ac

## Abstract

### Background

A common monogenic condition, β-thalassemia, is caused by a variety of mutations in the β-globin (*HBB*) gene. It is essential to accurately characterize these mutations for genetic counselling, diagnosis, and treatment.

### Objective

This study aimed to characterize and provide an updated and complete molecular spectrum of β-globin gene mutations, including both known and novel mutations, estimate their frequencies, and determine their possible deleterious effects.

### Methods

A total of 60 major β-thalassemia patients who sought treatment at Jeen Hospital in Duhok from August 2024 to February 2025 were analyzed using direct DNA sequencing of the β-globin gene.

### Results

Out of 60 sequence chromatograms, 40 were of excellent quality. Among these, 26 distinct mutations were found, comprising 10 exonic and 16 intronic variants. The most prevalent benign variants were IVSII-16 G>C (80%) and IVS II-666 C>T (77.5%), followed by Cd2 T>C [HBB:c.9 T>C, (62.5%)]. The pathogenic exonic mutations were found in coding regions, including Cd5 -CT [HBB:c.17_18delCT, (17.5%)], Cd6 A>T [HBB:c.20A>T (sickle cell mutation, 2.5%)], Cd8 A>G [HBB:c.26A>G (2.5%)], Cd39 C>T [HBB:c.118C>T, (5%)], Cd44 C>T [HBB:c.134C>T, (2.5%)], and Cd44 -C [HBB:c.135delC, (2.5%)]. Pathogenic intronic variants were also documented in splice

**Data availability statement:** All relevant data are within the manuscript and its Supporting information files.

**Funding:** The author(s) received no specific funding for this work.

**Competing interests:** No conflicting interests are disclosed by the authors.

junctions, including IVS I-1 G>A (15%) and IVS I-5 G>C (17.5%). Notably, seven novel variants were detected in this study, including four intronic variants (IVS I-129+C Ins, IVS II-72 G>A, IVS II-579 G>A, and IVS II-763+C Ins) and three exonic variants [Cd44 C>T (HBB:c.135C>T), Cd47 –G (HBB:c.142delG), and Cd118 –TT (HBB:c.355_356delTT)]. The majority of which were expected to be pathogenic or likely pathogenic based on variant location, predicted functional effect, and observed frequencies.

## Conclusion

The molecular investigation of β-thalassemia patients in Duhok showed a significant level of genetic variability in the β-globin gene and a high prevalence of compound heterozygosity among the β-thalassemia patients. The finding of several new variants is significant since it adds to the current mutation database and broadens the known mutational spectrum of the β-globin gene in this community. It also supports the necessity of thorough molecular diagnostics in regional management, screening, and genetic counseling of β-thalassemia.

## Introduction

β-thalassemia is a widespread anemia of a monogenic hereditary disorder with an autosomal recessive pattern of inheritance that occurs due to mutations in the beta-globin gene (*HBB*) [1]. The degree of β-globin chain imbalance is determined by the nature of the mutation of the β-globin gene. β0 refers to the complete absence of β-globin from the affected allele. β+ refers to alleles with some residual production of β-globin (around 10%). In β++, the reduction in β-globin production is very mild.

Clinically, β-thalassemia is linked to splenomegaly and bone marrow hyperplasia in addition to severe anemia [2]. At the clinical level, β-thalassemia can be classified into three primary categories: minor β-thalassemia (β/β+ or β/0), β-thalassemia intermedia (β0/β+), and β-thalassemia major (β0/β0) [3]. Two beta-globin gene abnormalities are the source of β-thalassemia major, which results in severe anemia and associated problems. Individuals identified as having β-thalassemia major require ongoing blood transfusions along with other medical interventions like iron chelating medications to prevent iron overload [4].

The human β-globin locus consists of five functional genes, *HBE*, *HBG2*, *HBG1*, *HBD*, and *HBB*, positioned on chromosome 11 in the same order in which they are expressed during ontogenesis. The β-Globin gene is found in the β-globin locus on the short arm of chromosome 11(11p15.4) and is composed of five genes (β-globin and beta-like genes span 70 kb). The β-Globin gene spans 1,600 bp, which encodes 146 amino acid residues. It consists of three exons and two introns (intervening sequences, IVS). Exon 1 encodes for amino acid residues from 1 to 29 and also encodes the first two bases of codon 30. Exon 2 encodes the last base of codon 30 and amino acids from 31 to 104. Exon 3 codes for amino acids from 105 to 146. In addition, 5' flanking, promoter, splicing, and 3' untranslated (3'-UTR) regions are also important for β-globin protein activity [5].

Presently, approximately 400 mutations in the beta-globin gene have been reported worldwide; the majority of them are single-nucleotide substitutions that result in faulty expression of the β-globin gene and lead to the disease's broad genotypic and phenotypic variability. Rarely, β-thalassemia results from gross gene deletion [6]. Although a handful of mutations constitute the bulk of beta-thalassemia mutations in each particular population, the distribution and frequency of mutations vary among ethnic groups and geographical locations [7]. Furthermore, studies are reporting new and uncommon mutations, suggesting that all regions need to be characterized for mutations, and the information about these mutations should be made public for prenatal diagnostics.

The World Health Organization estimated that approximately 68,000 people are born with β-thalassemia each year, and the incidence of β-thalassemia is estimated to be one in 100,000 worldwide [8]. High incidence has been reported in the Mediterranean, the Middle East, the Indian subcontinent, and Southeast Asia [3]. The high frequency of β-thalassemia among the Middle East population, particularly the Kurdish community, is thought to be largely caused by the traditional customs of consanguineous marriage and marriage within the same caste or ethnicity. With an average heterozygous carrier incidence of 4% and an estimated 15,000 registered individuals with thalassemia major and intermedia, β-thalassemia is a significant health issue in Iraq. With significant geographical differences in the occurrence of β-thalassemia mutations, hemoglobinopathies are serious health issues among Iraqi Kurds, with carrier rates ranging from 3.7% to 6.9% in the Kurdistan Region, Iraq. Of these, 27 distinct β-thalassemia mutations have been documented, based on published literature in this region of the world. The issue is made worse by the approximately 30% consanguineous marriage rate [7]. In this case, it would be imperative to start a preventive program for this illness. Establishing a program like this in a given group requires first identifying the mutations causing β-thalassemia in that population and building a database about the pattern and frequency of beta-globin mutations. Additionally, improved disease prevention would be made possible by identifying the uncommon and unique β-thalassemia variations.

Several studies have partially addressed the molecular basis of β-thalassemia in Duhok city, based on conventional methods of polymerase chain reaction and reverse hybridization of oligonucleotide probes corresponding to known β-thalassemia mutations [9]. However, aside from a single case-study report [10], to the best of our knowledge, the literature contains no previous research using direct DNA sequencing and mutation analysis of the whole β-globin gene. This would result in a knowledge deficit in the investigation of unexplored β-globin gene sites, as uncommon or new mutations are frequently missed by conventional techniques, which may lead to an incomplete understanding of the disease disorders, complicating diagnosis and treatment options. Thus, the current study aimed to provide a comprehensive molecular spectrum of β-globin gene mutations, define the unusual and/or novel mutations of hitherto unexplored sites of the β-globin gene in this area, estimate their frequencies, and determine their possible deleterious effects.

## Materials and methods

### Design and setting of the study

This cross-sectional study was carried out at the Thalassemia Disease Centre at Jeen Hospital in Duhok City, Kurdistan Region, Iraq.

### Inclusion and exclusion criteria

The inclusion criteria for transfusion-dependent thalassemia (TDT) were: age at initial presentation <2 years, Hb levels 6–7 g/dl, and regular transfusion requirement [11]. Patients who did not fulfill the criteria for diagnosis of TDT, as set by the current study, were excluded.

### Statement of ethics

The Research Ethics Committee (Ministry of Health, Duhok Directorate General of Health, and Ministry of Higher Education, Duhok Polytechnic University) gave its approval to this study. On (31/07/2024), the permission letter reference

number (31072024-6-26) was given. All patients or their guardians gave their written informed consent for the clinical specimens utilized in this investigation to be collected.

## Sample collection

The ethical board and scientific committee of Duhok Polytechnic University approved this study. This project involved 60 blood transfusion-dependent β-thalassemia (TDT) patients who registered at the Thalassemia Disease Center, Director-ate General of Health, Duhok city, Kurdistan Region-Iraq, and sought treatment and regular follow-up at Jeen Hospital in Duhok from August 2024 to February 2025. All enrolled patients were interviewed; demographic data, ethnicity, native language, consanguinity among parents, age, gender, and disease history were obtained. The medical records of the patients were obtained from the medical record information system and reviewed. The age at diagnosis, age at first trans-fusion, and frequency of transfusions were scrutinized. Informed consent was obtained from all patients or their guardians. Blood samples were accessed and collected with EDTA anticoagulants from 5th August 2024–27th February 2025, for downstream laboratory examinations, DNA extraction, and genetic analysis [11,12].

## DNA extraction

The DNA was extracted from the blood using the Presto™ Mini gDNA Extraction Kit (Geneaid/Taiwan) following the man-ufacturer's instructions. NanoDrope2000 spectrophotometer (Applied Biosystem) was used to assess the extracted DNA's concentration and purity.

## PCR amplification of the *HBB* gene

All DNA samples underwent *in vitro* amplification by polymerase chain reaction (PCR) to amplify the entire beta-globin gene, encompassing three exons and two introns, using two sets of primers (Table 1) as described previously [13]. The typical 20µl PCR reaction mixture was formulated, comprising 1X TaqMaster mix (Promega), 10 pmol of forward primer (1µl), 10 pmol of reverse primer (1µl), and 20 ng of template DNA. The usual PCR settings for set I primers comprised an initial denaturation at 95°C for 3 minutes, followed by 30 cycles of 30 seconds at 95°C, 30 seconds at 58°C (annealing temperature), and 30 seconds at 72°C (extension temperature), concluding with a final extension of 3 minutes at 72°C. The thermal cycling of set II primers was conducted under conditions analogous to those of set I primers. The amplified products were stained using safe dye staining following electrophoresis on a 1.5% agarose gel in 1X TAE buffer, and the gel was electrophoresed at 75 volts for 45 minutes for verification and confirmation. The bands were visualized on a UV-transilluminator and captured using a photographic recording method (GBOX) [3].

## Direct sequencing of the *HBB* gene

Following the guidelines provided by the manufacturer, the PCR products of the *HBB* gene were purified using a MinElute PCR Purification Kit (50) (Qiagen). Macrogen Inc. (Seoul, South Korea) employed an automated ABI3730XL genetic

Table 1. The set of primers used to amplify and sequence the beta-globin gene.

| Gene | Primer Name | Sequence 5'-3' | Position* | Amplicon size |
|---|---|---|---|---|
| B-Globin gene (*HBB*) | HBB1-F | AGAAGAGCCAAGGACAGGTACG | 61991−61990 | 760 bp |
| | HBB1-R | TGCAATCATTCGTCTGTTTCCC | 62730-62751 | |
| | HBB2-F | TCCCTAATCTCTTTCTTTCAGG | 63190-63211 | 660 bp |
| | HBB2-R | TTTTCCCAAGGTTTGAACTAGC | 63829-63849 | |

*Positions based on GenBank accession number U01317.

analyzer for subsequent Sanger sequencing of the purified PCR products. The primer pair of HBB1 and HBB2 was used separately in individual reactions for the bidirectional sequencing of the whole *HBB* gene using each of the forward and reverse primers, independently.

## Data analyses

Sequencing data were submitted to the GenBank using the BLAST method in the National Center for Biotechnology Information (NCBI) database to confirm the identity of the patients' sequences. The patients' sequences were compared with the NCBI Reference sequence (NG_000007.3, Homo sapiens chromosome 11, GRCh38.p13 Primary Assembly) using the Multalin alignment tool. Mutation frequency was determined by direct counting. Ithanet, HbVar, dbSNP IDs, and ClinVar databases were used for the identification and description of the mutations that had been identified in other populations [14]. A variant was considered novel if it was absent in all these major variant databases and population studies and had not been described in peer-reviewed literature. To calculate the probability of deleterious effects of novel mutation, different bioinformatics tools available online were used, viz EXPASY Translate tool (https://web.expasy.org/translate/) to translate the DNA sequence to protein sequence. The ACMG/AMP 2015 criteria were used for the pathogenicity classification of novel variants as pathogenic/likely pathogenic following the PVS1 (null variant in a gene where loss of function is a known disease mechanism) and PM2 (absence from population databases) criteria [15].

## Results

In the current study, a complete spectrum of β-globin gene (*HBB*) mutations was detected and reported. Of the 60 enrolled patients, 40 yielded high-quality, analyzable DNA sequence chromatograms. Out of them, 26 distinct mutations were identified, comprising 16 intronic and 10 exonic variants. Remarkably, seven of these variants were novel, reported here for the first time worldwide, and had never been reported in any of the pertinent databases, including dbSNP IDs, HGVS nomenclature, IthaGenes, HbVar, and ClinVar. Each novel mutation exhibits unique clinical significance, molecular effects, and different frequencies within the patient cohort (Table 2). The frequency, chromosomal location, and clinical relevance of each variation were determined using the current variant interpretation and classification guidelines (ACMG/AMP) and the aforementioned databases.

Starting with the intronic variants, a total of 16 intronic mutations were found in the β-globin gene's introns 1 (IVS-I) and 2 (IVS-II). Intron 1 contained seven variants that mostly affected the splice donor and splice region: IVS I-5 G > C (HBB:c.92 + 5G > C, rs33915217) was the most common, occurring in 17.5% of samples, followed by IVS I-1 G > A (HBB:c.92 + 1G > A, rs33971440) and IVS I-6 T > C (HBB:c.92 + 6T > C, rs35724775) observed in 15% and 10% of cases, respectively. These substitutions are known to disrupt normal splicing and are considered pathogenic. One case (2.5%) contained a novel insertion, IVS I-129 + C Ins (HBB:c.92 + 129_92 + 130insC), which was deemed likely benign because it was located in the deep region of intron 1 (Fig 1A).

Additionally, Intron 2 was found to have nine variations. Of which, IVS II-16 G > C (rs10768683) was the most prevalent variant overall, occurring in 80% of patients, yet being regarded as benign. IVS II-666 (C > T) (rs1609812) was also identified in 77.5% of instances and is categorized as benign. Though less common, other variations, including IVS II-1 G > A (10%) and IVS II-763 + C Ins (5%), which have the potential to be pathogenic, were also present. The discovery of three new intronic variants, including IVS II-72 G > A (HBB:c.315 + 72 G > A) (Fig 1B), IVS II-579 G > A (HBB:c.316−272 G > A) (Fig 1C), and IVS II-763 + C Ins (HBB:c.316−87_ 316−88insC) (Fig 1D), suggests even more variation in non-coding areas.

In the gene's coding regions (exons 1–3), ten genetic variations were found. These included a variety of mutation types, such as frameshift, missense, nonsense, and silent (synonymous) mutations. The prevalence of these variations in the research samples and their anticipated effects on protein function differ. There were four different variations found in exon 1. A silent alteration at Cd2 CAT > CAC (HBB:c.9T > C) was the most common, occurring in 62.5% of samples and deemed

**Table 2. *HBB* Gene Variants Found in the Study Cohort (n = 40) distributed across intronic and exonic regions.**

| Common Name | Transcript (NM_000518.5) and HGVS name | dbSNP Identifier | Genomic Locus (NG_000007.3) | Variant Type | Frequency (%) | ACMG/AMP Classification |
|---|---|---|---|---|---|---|
| IVS I-1 G>A | HBB:c.92+1G>A | rs33971440 | g.70687G>A | Splice donor substitution | 6/40 (15%) | Pathogenic |
| IVS I-5 (G>C) | HBB:c.92+5G>C | rs33915217 | g.70691G>C | Splice region substitution | 7/40 (17.5%) | Pathogenic |
| IVS I-5 (G>T) | HBB:c.92+5G>T | rs33915217 | g.70691G>T | Splice region substitution | 1/40 (2.5%) | Pathogenic |
| IVS I-6 (T>C) | HBB:c.92+6T>C | rs35724775 | g.70692T>C | Splice region substitution | 4/40 (10%) | Pathogenic |
| IVS I-6 (T>G) | HBB:c.92+6T>G | rs35724775 | g.70692T>G | Splice region substitution | 1/40 (2.5%) | Pathogenic |
| IVS I-110 G>A | HBB:c.93-21G>A | rs35004220 | g.70796G>A | Intronic substitution | 2/40 (5%) | Pathogenic |
| IVS I-129+C Ins | HBB:c.92+129_92+130insC | Novel | g.70815_70816insC | Deep intronic insertion variant | 1/40 (2.5%) | Likely Benign |
| IVS II-1 G>A | HBB:c.315+1G>A | rs33945777 | g.71040G>A | Splice donor substitution | 4/40 (10%) | Pathogenic |
| IVS II-16 G>C | HBB:c.315+16G>C | rs10768683 | g.71055G>C | Intronic substitution | 32/40 (80%) | Benign |
| IVS II-72 G>A | HBB:c.315+72 G>A | Novel | g.71111G>A | Intronic substitution variant | 1/40 (2.5%) | VUS/Uncertain |
| IVS II-74 T>G | HBB:c.315+74T>G | rs7480526 | g.71113T>G | Intronic substitution variant | 3/40 (7.5%) | Benign |
| IVS II-81 (C>T) | HBB:c.315+81C>T | rs7946748 | g.71120C>T | Intronic substitution variant | 4/40 (10%) | Benign |
| IVS II-100 G>A | HBB:c.315+100G>A | rs899975457 | g.71139G>A | Intronic substitution variant | 1/40 (2.5%) | Benign |
| IVS II-579 G>A | HBB:c.316−272 G>A | Novel | g.71792G>A | Intronic substitution variant | 1/40 (2.5%) | Benign |
| IVS II-666 (C>T) | HBB:c.316-185C>T | rs1609812 | g.71705C>T | Intronic substitution variant | 31/40 (77.5%) | Benign |
| IVS II-763+C Ins | HBB:c.316−87_ 316−88insC | Novel | g.71607_71608insC | Intronic insertion variant | 2/40 (5%) | Likely pathogenic |
| Cd2 CAT>CAC | HBB:c.9 T>C | rs713040 | g.70603T>C | silent substitution variant | 25/40 (62.5%) | Benign |
| Cd5 -CT | HBB:c.17_18delCT | rs63750729 | g.70611_70612del | Frameshift deletion | 7/40 (17.5%) | Pathogenic |
| Cd6 GAG>GTG | HBB:c.20A>T | rs334 | g.70614A>T | Missense variant (Glu>Val) | 1/40 (2.5%) | Pathogenic (sickle cell disease) |
| Cd8 AAG>AGG | HBB:c.26A>G | rs33932981 | g.70620A>G | Missense variant (Lys>Arg) | 1/40 (2.5%) | Pathogenic |
| Cd39 CAG>TAG | HBB:c.118C>T | rs11549407 | g.70842C>T | Nonsense variant (Gln>STOP) | 2/40 (5%) | Pathogenic |
| Cd44 TCC>TTC | HBB:c.134C>T | -- | g.70858C>T | Missense variant (Ser>Phe) | 1/40 (2.5%) | Pathogenic |
| Cd44 TCC>TCT | HBB:c.135C>T | Novel | g.70859C>T | silent substitution variant (Ser>Ser) | 1/40 (2.5%) | Benign |
| Cd44 -C | HBB:c.135delC | rs80356820 | g.70859del | Frameshift deletion | 1/40 (2.5%) | Pathogenic |
| Cd47 -G | HBB:c.142delG | Novel | g.70866delG | Frameshift deletion | 1/40 (2.5%) | Pathogenic |
| Cd118 -TT | HBB:c.355_356delTT | Novel | g.71929_71930delTT. | Frameshift deletion | 1/40 (2.5%) | Pathogenic |

Abbreviations: HGVS: Human Genome Variation Society, dbSNP: Database for Single Nucleotide Polymorphisms.

Note: All variants are reported using HGVS nomenclature with RefSeq transcript NM_000518.5 and RefSeqGene NG_000007.3 on GRCh38.p13. The ACMG/AMP 2015 classification is based on predicted pathogenicity. The known variations were evaluated using the existing variation field (dbSNP rsIDs) and corroborated with ClinVar and GnomAD.

benign. Missense mutations like Cd6 GAG>GTG (HBB:c.20A>T, p.Glu6Val), the classical cause of sickle cell disease (HbSS) mutation, and frameshift deletions like Cd5-CT (HBB:c.17_18delCT) were found at lower frequency (2.5–17.5%) and were deemed pathogenic because of their disruptive effects on protein structure and function.

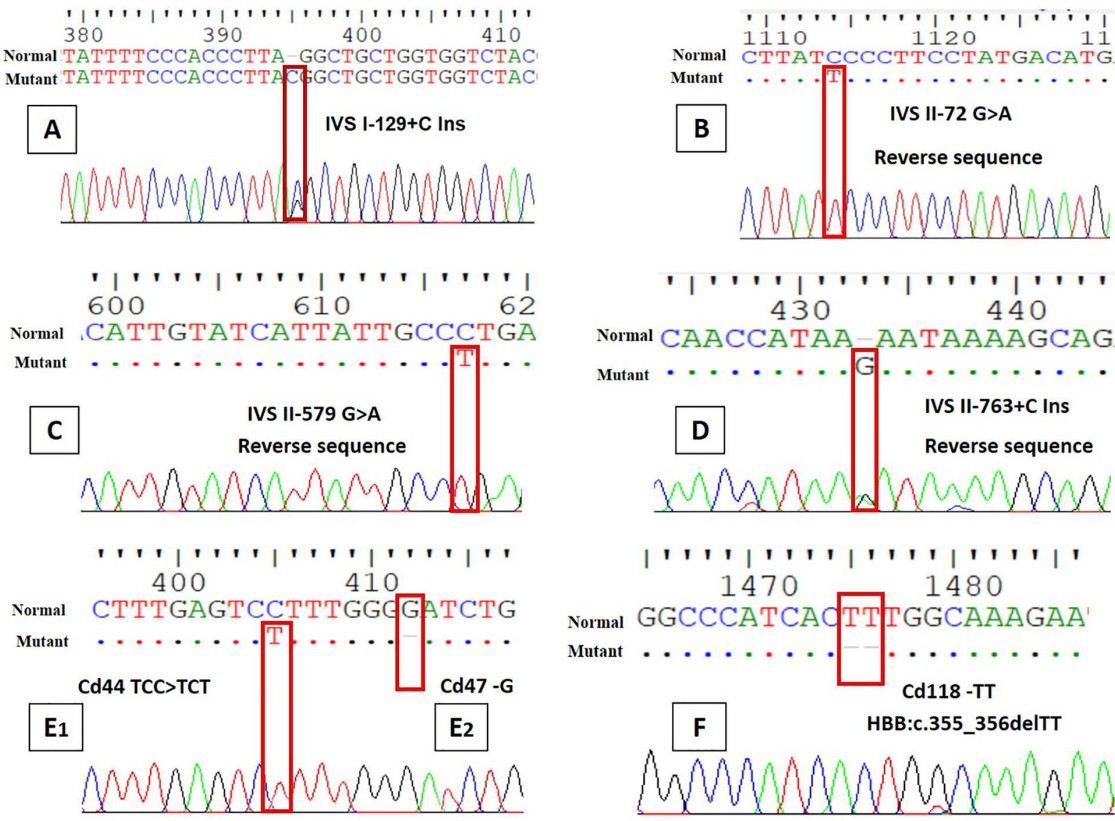

**Fig 1. Chromatograms of the novel *HBB* gene mutations identified in this study.**

Within exon 2, five different variations were found. Notably, 5% of the samples had the Cd39 CAG > TAG (HBB:c.118C > T), a nonsense mutation that introduced a premature stop codon to the β-globin chain. Remarkably, codon 44 showed a range of disease-causing mutations among the studied cohort, including a frameshift deletion (HBB:c.135delC) and missense alterations (HBB:c.134C > T) in addition to a novel synonymous (silent) variation (Cd44 TCC > TCT; HBB:c.135C > T). Additionally, another novel frameshift deletion (Cd47 –G; HBB:c.142delG) was detected in 2.5% of the samples, which was deemed as pathogenic.

Furthermore, a new frameshift deletion mutation in exon 3, called Cd118-TT (HBB:c.355_356delTT), which disrupts the reading frame, was identified in 2.5% of patients. This mutation likely results in a shortened and inoperable β-globin protein. It is classified as pathogenic because of its expected impact on the structure and function of the β-globin protein.

The identification and reporting of seven unique mutations, out of the 26 discovered variants, that had not been previously reported in public databases like dbSNP, ClinVar, HbVar, and other pertinent databases, is remarkable. The *HBB* gene has three new intronic variations, all of which were found in non-coding areas that may have an impact on gene regulation, RNA splicing, or transcript stability. A single-nucleotide cytosine insertion in intron 1 called IVS I-129 + C Ins (HBB:c.92 + 129_92 + 130insC) is located close to regulatory elements and has the potential to interfere with splicing enhancer or silencer sites. Intron 2 has the guanine-to-adenine alteration known as IVS II-72 G > A (HBB:c.315 + 72G > A), which is situated 72 nucleotides downstream of exon 2. This area could be important for identifying intron-exon boundaries. Intron 2 has the guanine-to-adenine transition IVS II-579 G > A (HBB:c.316-272G > A), which is located 272 nucleotides upstream of exon 3 and may have an impact on intronic regulatory motifs. Specifically, IVS II-763 + C Ins and IVS

I-129 + C Ins were deemed to be probable pathogenic and require further research. Furthermore, four new variations were found in the exonic areas. A silent mutation at codon 44, Cd44 TCC > TCT (HBB:c.135C > T) (Fig 1E$_1$), may affect codon use or mRNA stability but does not change the amino acid (Ser > Ser). The β-globin chain is expected to be shortened and rendered inoperable by the frameshift deletion at codon 47 known as Cd47 -G (HBB:c.142delG) (Fig 1E$_2$). The pathogenic mutation of Cd118 -TT (HBB:c.355_356delTT) (Fig 1F) was another frameshift loss in exon 3. The disruptive character of this two-nucleotide loss frameshift deletion and its anticipated effect on protein synthesis led to its classification as pathogenic.

In this study, a total of 23 different genotypes were identified among the 40 β-thalassemia patients analyzed, which indicated a diverse mutation spectrum. The genotypic pattern showed that 92.5% (37/40) of the investigated β-thalassemia subjects were compound heterozygous for two or more different mutations (S1 Table), suggesting that complex compound-heterozygous genotypes are highly prevalent. Within this 40-patient cohort, the *HBB* genotype burden varied from 1 to 5 different variations per patient (mean ± SD = 3.60 ± 1.20; median = 4). Of these, 10% (4/40) of patients carried two different mutations, 25% (10/40) had three, and 27.5% (11/40) had five distinct variants. Additionally, the most frequent group, 30% (12/40), had four various mutations. In contrast, simple homozygosity was uncommon, in which only 3 patients (7.5%) were homozygous for a single pathogenic variant, of which two cases had IVS I-5 G > C and one case had Cd8 A > G genotype. These findings were consistent with transfusion-dependent β-thalassemia and demonstrate a high frequency of compound heterozygosity in the studied population. Furthermore, among the compound heterozygous profiles, the most prevalent transfusion-dependent genotype was (Cd2 T > C, Cd5 –CT, IVS II-16 G > C, IVS II-666 C > T) observed in 15% of the studied cohort, accompanied by (Cd2 T > C, IVS II-16 G > C, IVS II-666 C > T) in 10%, and each of (Cd2 T > C, IVS I-110 G >A, IVS II-16 G > C, IVS II-74 T > G, IVS II-666 C > T) and (Cd2 T > C, IVS I-1 G >A, IVS II-16 G > C, IVS II-666 C > T) in 7.5%, respectively, indicating recurrent combinations. Other less frequent genotypic combinations (5% frequency) were (Cd2 T > C, IVS I-1 G >A, IVS II-1 G >A, IVS II-16 G > C, IVS II-666 C > T), (IVS I-6 T > C, IVS II-16 G > C, IVS II-666 C > T), and (IVS II-16 G > C, IVS II-666 C > T). However, 40% of the identified genotypes, which ranged from a single homozygous variant to five different variants in compound heterozygosity, were unique to single patients. Additionally, 17.5% of patients (7/40) carried at least one novel variant in the dataset. Altogether, these findings showed a significant mutation burden per patient with notable allelic heterogeneity.

## Discussions

The current study broadens the genetic epidemiology of β-thalassemia in Iraq by identifying 26 different HBB variations in a patient cohort under study, including seven previously unknown alterations. The global pattern of a relatively small core of highly prevalent mutations accompanied by a long "tail" of population- or family-specific changes is reflected in this breadth of allelic diversity. According to a recent synthesis, over 350 β-thalassemia alleles have been described worldwide, but only about 20 of these account for more than 80% of cases [16]. Therefore, our findings support the idea of significant regional heterogeneity and highlight the need for region-specific mutation panels for prenatal diagnosis and carrier identification.

In our dataset, three classical splice-site mutations (IVS I-5 G > C, IVS I-1 G >A, and IVS I-6 T > C) together accounted for 42.5% of the pathogenic alleles. These lesions have been reported to be similarly prevalent in the Iraqi premarital screening program (IVS I-5 G > C = 12.4%; IVS I-6 T > C = 8.3%), as well as in a previous Baghdad study when IVS I-5 G > C came in third rank overall [17]. On a regional level, IVS I-5 G > C is also the most prevalent allele in Bangladesh (81%) [18], Pakistan (44%) [19], and portions of Southeast Asia and Thailand (34%–40%) [4,20], indicating a South-to-West trend that most likely reflects historical gene flow along trade routes. Transfusion-dependent symptoms in our cohort are explained by these changes, which result in β⁰- or severe β⁺-thalassemia transcripts by eliminating or weakening canonical splice donor sites.

Conversely, the most common alleles overall were IVS II-16 G>C (80%) and IVS II-666 C>T (77.5%), both of which are regarded as benign polymorphisms. Similar elevated carriage rates (60 percent for IVS II-16 G>C) have been seen in healthy Bangladeshi controls [18]. However, there is now emerging evidence that these variations can regulate illness when co-inherited with severe alleles or Hb S. Case reports demonstrate the change of basic sickle-cell trait to Hb S/β-thalassemia intermedia when either polymorphism is present [21,22]. While the transfusion reliance of participants is mostly explained by the presence of severe β° or splice-abolishing alleles, the occurrence of supposed benign polymorphisms (e.g., IVS II-16 G>C) may operate as disease modifiers by changing the α/β-globin imbalance or residual β-chain production. Deep-intron variations in other populations have been suggested to have similar modifying functions [22]. Although the current study may not have been powered to analyze such epistatic interactions, the extremely high background frequency necessitates caution when interpreting compound genotypes in clinical settings.

We found three intronic alterations that had not been reported before: IVS II-129 insC, IVS II-72 G>A, and IVS II-579 G>A. All of them are located in deep-intron areas that are enriched for cryptic splice enhancers and silencers. This suggests that they may disrupt or generate binding sites for hnRNP A1 or SR-protein, and as such, they should be studied using functional minigene splicing tests. The ACMG/AMP guidelines [15,23] support the preliminary "likely pathogenic" diagnosis of IVS II-763 insC, a fourth new variation that was found in 5% of patients and consistently co-segregated with severe anemia.

In the current investigation, ten exonic mutations have been reported in the *HBB* gene. Collectively, they ranged from subtle variations that have little to no effect on phenotype to severe mutations that cause β-thalassemia or sickle cell disease. With worldwide allele frequencies as high as 0.8%, the silent c.9 T>C (codon 2) variant reached 62.5%; ClinVar and gnomAD classify this variant as benign [15,24]. Its significant local incidence highlights the need to avoid incorrectly categorizing synonymous alterations as disease-causing in diagnostic reports and implies linkage disequilibrium with a Middle-Eastern *HBB* haplotype [25].

The study revealed that 17.5% of the participants had the famous HBB:c.20A>T [Glu6Val (Hb S)] mutation, highlighting the persistent gene flow between the northern Iraqi β-thalassemia and sickle-cell endemic areas. Intermarriages, migrations, and demographic exchanges between tribes living in these overlapping genetic belts are all reflected in this discovery. For the therapy of hemoglobinopathies in the area, the comparatively high frequency of Hb S points to a considerable mixing that might have both clinical and epidemiological consequences. With almost 5% of the cases, the codon 39 (CAG>TAG) nonsense mutation was found to be the second most common coding lesion. Its well-established position as the second most prevalent β-thalassemia allele in the larger Mediterranean basin is in line with this. Interestingly, in Sardinia and several regions of North Africa, this mutation is also the most common form that causes β-thalassemia [26,27]. Consistent with earlier research carried out in Baghdad [17], the current cohort once more exhibited the characteristic frameshift mutations Cd5 delCT and Cd44 delC. The continuous reporting of these recurrent mutations in previous regional studies suggests that they may be enriched in the local population. Three new mutations were found in addition to these known variations: a silent substitution at Cd44 (c.135C>T) and frameshifts Cd47 delG and Cd118 delTT. Premature termination codons are expected to be introduced by all these pathogenic mutations, most likely leading to nonsense-mediated mRNA decay (NMD). A β°-thalassemia phenotype, which is defined by the total lack of functional β-globin production, is effectively caused by this degradation process, which stops the synthesis of shortened β-globin chains. This discovery highlights the significance of thorough molecular screening in genetic diagnosis and counselling and adds to the growing mutational spectrum of β-thalassemia in the Iraqi population, especially the new variations.

Through full HBB locus sequencing, we identified 26 unique variations, including seven entirely new alterations in a current cohort from Duhok, Kurdistan Region, Iraq. The regional allele inventory now has over 40 mutations, which is wider than any single-center Kurdish research to date. Just 11–22 mutations were usually detected in earlier studies that relied on strip hybridization or restricted Sanger panels [9]. Thus, our findings highlight the need to update routine diagnostic panels and the discovery capability of full-gene methods.

The IVS-II-1 G>A and IVS-I-6 T>C continue to be the dominant pathogenic alleles across all Kurdish provinces. They accounted for 25% of the mutations in the present cohort and for 24% to 43% of the mutations in the successive Duhok studies from 2006 to 2021 [9,12], 39% of the transfusion-dependent patients from Dohuk/Erbil [28], and 54% of the β-thalassemia inter-media alleles [29]. Notwithstanding methodological and phenotypic variations, the 2020 Sulaymaniyah audit verified the same hierarchy (IVS-II-1=35.7%, IVS-I-6=18%) [11]. Strong founder effects that are strengthened by consanguinity are supported by this temporal stability. However, this iconic image has been altered by two new datasets. In comparison to 2006 data, the 2021 "updated spectrum" research revealed that the influx of Yazidi and other displaced families roughly increased the prevalence of IVS-I-110 G>A and codon8 -AA [12]. Additionally, we found a little increase in these alleles (4–6%), which points to ongoing gene flow in the Mosul–Duhok corridor. And, after comprehensive sequencing took the place of strip testing, a 2023 Sanger-sequencing survey found that 59% of patients had IVS-II-666 C>T and IVS-II-16 G>C, along with five new deep-intron alter-ations [3]. This result is consistent with our own dataset (77–80%), demonstrating how previously "invisible" polymorphisms become apparent when the full gene is examined. Furthermore, the β-thalassemia-intermedia cohort, where the moderate IVS-I-6 T>C allele alone explained one-third of cases [29], contrasts with transfusion-dependent cohorts, which are more likely to have severe β° or splice-abolishing alleles, for example, 88% of variants explained by seven mutations in the 2010 series [28].

Our study adds seven new variations to the Kurdish mutation pool that had not been reported before. Notably, we uncovered a codon 118 delTT frameshift that independently supports a 2021 solitary case account [10], albeit with a different position at the same codon (355_356 vs. 356_357). Both result in a frameshift at codon 118 and most likely produce the same shortened/nonfunctional β-globin chain. This confirms that this β° allele is a low-frequency founder mutation rather than an anecdote. Five further unique intronic insertions and deletions were included in the Sulaymaniyah 2023 investigation [3], highlighting the fact that non-coding areas are still understudied sources of pathogenic alteration. Although it still targets the 23 "common" mutations, the Iraqi premarital kit, which has been frozen since 2010, would miss 29% of the alleles in our cohort and 41% of the most recent Sulaymaniyah series. First-tier sensitivity would increase to >90% with the addition of IVS-I-110 G>A, codon8 -AA, high-frequency intronic SNPs, and the seven new pathogenic mutations reported here. Reflex next-generation sequencing, which has already been implemented in a few of the UK screening hubs, would future-proof detection in the face of ongoing migration and mutational drift when resources allow.

The high frequency of compound heterozygosity in β-thalassemia major was consistent with transfusion-dependent β-thalassemia in the studied population. Additionally, the findings pointed to compound heterozygosity as the primary cause of the observed genetic architecture and phenotypic heterogeneity, including transfusion dependency and severity levels. The high level of compound heterozygosity reported in the current population aligns with previous studies [30,31], which also reported a similar predominance of compound heterozygous genotypes in other β-thalassemia cohorts. This may suggest a multifactorial inheritance pattern, in which several mutations contribute to the disease manifestation. Additionally, the recurrent genotypic combinations reported in this study suggest shared ancestry, or the population may have experi-enced founder effects. The presence of these recurrent mutations in different groups suggests that they may contribute to the pathophysiology of β-thalassemia [32]. Furthermore, the study population's genetic diversity was demonstrated by the 37.5% of patients who had unique genotypes found. Other investigations have reported similar findings, with a sizable per-centage of patients displaying distinct genotypic profiles [33]. Overall, the genotypic clustering and zygosity pattern findings of this study demonstrated the importance of precise genotyping for genetic counseling, diagnosis, and prognosis.

## Conclusions

Altogether, by identifying a diverse range of *HBB* variations, including 17 (65%) variants categorized as pathogenic or poten-tially pathogenic and several benign or unique polymorphisms, such as deep-intron and frameshift events, this work expands the Iraqi β-thalassemia mutation database. The detection of a remarkably diverse mutation spectrum and high prevalence of compound heterozygosity among the β-thalassemia patients is indicative of intricate allelic interactions that lead to the severe, transfusion-dependent clinical phenotypes observed. The development of locus-specific therapies tailored to the allelic

landscape of Iraq should be guided by these insights, which should be incorporated into regional diagnostic algorithms to enhance carrier detection and reproductive risk assessment. To establish the identified new mutations' clinical significance and ascertain their specific biological impact, further functional validation and segregation analysis are recommended.

## Supporting information

**S1 Table. Supporting information.**
(DOCX)

## Acknowledgments

We express our gratitude to the Thalassaemia Disease Centre and the Duhok Central Laboratory of the Directorate General of Health in Duhok, Kurdistan Region-Iraq, for their invaluable assistance and provision of all facilities.

## Author contributions

**Conceptualization:** Hemin Esmael Othman.

**Data curation:** Hemin Esmael Othman.

**Formal analysis:** Hemin Esmael Othman.

**Investigation:** Torin Abdulaziz Sadoon.

**Methodology:** Torin Abdulaziz Sadoon.

**Resources:** Torin Abdulaziz Sadoon.

**Software:** Hemin Esmael Othman.

**Supervision:** Hemin Esmael Othman.

**Writing – original draft:** Torin Abdulaziz Sadoon.

**Writing – review & editing:** Hemin Esmael Othman.

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
