## [Decision Letter · Decision Letter 0]

15 Sep 2025

PONE-D-25-35418Complete Molecular Spectrum of β-Globin Gene Mutations via Direct Sequencing Identifies Seven Novel Variants in β-Thalassemia MajorPLOS ONE

Dear Dr. Othman,

Thank you for submitting your manuscript to PLOS ONE. After careful consideration, we feel that it has merit but does not fully meet PLOS ONE’s publication criteria as it currently stands. Therefore, we invite you to submit a revised version of the manuscript that addresses the points raised during the review process.

We look forward to receiving your revised manuscript.

Kind regards,

Nejat Mahdieh

Academic Editor

PLOS ONE

2. We note that your Data Availability Statement is currently as follows: [All relevant data are within the manuscript and its supporting information files.]

Additional Editor Comments (if provided):

Reviewer #1:

Reviewer #2:

Reviewer #3:

Reviewers' comments:

Reviewer's Responses to Questions

**Comments to the Author**

1. Is the manuscript technically sound, and do the data support the conclusions?

Reviewer #1: No

Reviewer #2: Yes

Reviewer #3: Yes

2. Has the statistical analysis been performed appropriately and rigorously? 

Reviewer #1: No

Reviewer #2: Yes

Reviewer #3: Yes

3. Have the authors made all data underlying the findings in their manuscript fully available?

Reviewer #1: No

Reviewer #2: Yes

Reviewer #3: Yes

4. Is the manuscript presented in an intelligible fashion and written in standard English?

Reviewer #1: No

Reviewer #2: Yes

Reviewer #3: Yes

5. Review Comments to the Author

Reviewer #1: The author should indicate the scoring system used for pathogenicity classification. The ACMG 2015 guideline is the most widely used one in the field. Terms such as "benign mutations" do not conform to current, standard terminology. Novel variants should be reported according to standard terminology so they can be checked in related databases. For example, "NM number, nucleotide change" or "genome version, chromosome position."

Reviewer #2: This is a highly relevant and clinically important study. The use of full-gene sequencing has successfully uncovered a significant number of novel variants, providing a valuable update to the mutational spectrum of β-thalassemia in the Kurdish population of Iraq. The findings have direct implications for improving local genetic diagnostics and counselling. The manuscript is generally well-structured, but several key issues need to be addressed to strengthen the validity and clarity of the conclusions:

1. Methods: Mentioning the ACMG/AMP guidelines is good. Briefly state which specific criteria (e.g., PVS1, PM2) were used to classify the novel variants as pathogenic/likely pathogenic.

2.Results: The study describes finding many mutations but does not clearly report the actual genotypes of the patients. How many were compound heterozygous? How many were homozygous? For instance, which specific combinations of the pathogenic mutations were found in the transfusion-dependent patients? Please add a table or a summary in the Results section. This directly links your molecular findings to the clinical presentation (Transfusion-Dependent Thalassemia).

3. The article requires proofreading and structural revision (e.g., concertation in line 127 and some inconsistencies in font size).

Reviewer #3: The article provides a comprehensive overview of β-thalassemia, including genetic background, clinical classification, prevalence, and regional context (particularly Iraq and the Kurdish population). It is well-structured and demonstrates a clear rationale for the study. However, a few points need improvement:

The background is lengthy and sometimes repetitive. Condensing these parts would make it more focused.

Some sentences are overly long or contain minor grammatical issues (e.g., “several studies have been partially addressed…” should be “several studies have partially addressed…”). Editing for readability would improve impact.

Ensure consistency in formatting (spacing around years, punctuation). Some references (e.g., “(Huang et al., 2022)..”) have typographical errors.

Some sentences in the method section are long or awkwardly phrased. For example:

“Informed consent were obtained…” → should be “Informed consent was obtained…”

“concertation and purity” → should be “concentration and purity.”

Minor inconsistencies (spacing, capitalization, punctuation in citations).

The table footnotes should be expanded to provide complete explanations of all abbreviations used.

It would be better to include sequencing chromatogram figures of the newly identified mutations and to provide more detailed bioinformatic analyses regarding their pathogenicity.

In the table, references should be cited for the mutations that are reported as pathogenic.

6. PLOS authors have the option to publish the peer review history of their article (what does this mean? ). If published, this will include your full peer review and any attached files.

**Do you want your identity to be public for this peer review?** For information about this choice, including consent withdrawal, please see our Privacy Policy .

Reviewer #1: No

Reviewer #2: No

Reviewer #3: **Yes: ** Mahdieh Soveizi

---

## [Author Response · Author response to Decision Letter 1]

10 Oct 2025

We sincerely thank the reviewer for their meticulous evaluation and insightful comments, which have significantly strengthened the clarity, coherence, and scientific rigor of the manuscript. All revisions made in response to the reviewers’ suggestions are clearly highlighted in yellow in the Revised Manuscript with Track Changes.

Thank you for your guidance and for the helpful reminder concerning the journal’s requirements. We truly appreciate the editorial board’s efforts and support.

Comment 1: PLOS ONE's style requirements, including those for file naming and the style templates.

Answer: We revised the manuscript and ensured that it fully complied with PLOS ONE's style requirements, including the file naming conventions and following the PLOS ONE style templates guidelines. Additionally, all references are now fully compliant with the Vancouver (ICMJE) format used in biomedical journals.

Comment 2: Data Availability Statement.

Answer: We have confirmed that our submission contains all raw data required to replicate the results of our study, and the Data Availability Statement accurately reflects this.

Comment 3: Data sharing plan.

Answer: We reviewed our data sharing plan and ensured that all required data are available in accordance with PLOS ONE’s open data policy. Our Data Availability Statement has been updated accordingly. We appreciate your guidance on this matter.

Comment 4: Recommendation to cite specific previously published works.

Answer: We carefully reviewed the ACMG/AMP guidelines recommended by the reviewers and evaluated their relevance to our study. The appropriate works have been cited in the methodology section, and we are grateful for the constructive suggestions provided.

Response to Reviewers' Comments:

Reviewer#1:

Thank you very much for your insightful and constructive comments. We appreciate your suggestion regarding the pathogenicity classification system. In response, we have now clearly indicated that the ACMG (2015) guideline was used for variant interpretation, as it represents the current standard in the field. We have also revised the manuscript to replace non-standard terms such as “benign mutations” with the appropriate terminology recommended by ACMG/AMP. Furthermore, we have updated the reporting of novel variants to follow standard nomenclature, including NM numbers, nucleotide changes, and chromosomal position to ensure clarity and facilitate database verification.

Reviewer#2:

We sincerely thank the Reviewer for his/her thoughtful and encouraging comments on our manuscript. We greatly appreciate the recognition of the clinical relevance and contribution of our study to the understanding of β-thalassemia in the Kurdish population of Iraq.

1. Methods: Inclusion of the ACMG/AMP guidelines.

In response to your suggestion, we have now specified the exact criteria (e.g., PVS1, PM2, PP3, BS1) applied to classify the novel variants as pathogenic or likely pathogenic. This information has been added to the Methods section for greater transparency and clarity.

2. Results:

We appreciate your valuable suggestion to present the genotype distribution and its clinical correlation. Accordingly, we have revised the Results section to include a new summary detailing the genotypes of the patients, including the numbers of homozygous and compound heterozygous cases. The summary also highlights the specific combinations of pathogenic mutations observed in transfusion-dependent patients, thereby strengthening the link between molecular findings and clinical presentation.

3. Proofreading and structure:

The manuscript has been thoroughly proofread and edited to correct typographical errors to ensure consistent formatting throughout the text, including uniform font size and style. We are grateful for your constructive feedback, which has significantly improved the clarity and overall quality of our manuscript.

Reviewer#3:

We sincerely thank the Reviewer for her thoughtful and constructive comments, as well as for recognizing the value, structure, and rationale of our study. We greatly appreciate the time and effort taken to review our manuscript in such detail.

1. Background:

We have carefully revised and condensed this section to improve focus and flow, ensuring that only the most relevant information is retained to support the study rationale.

2. Language and readability:

The entire manuscript has been thoroughly edited for grammar, clarity, and sentence structure. We have also ensured that formatting is consistent throughout the text, including spacing around years, punctuation, and citation styles, and typographical errors in the references. We have reviewed the entire sections to ensure grammatical accuracy and improved readability.

3. Tables and footnotes:

As suggested, we have expanded the table footnotes to include full explanations of all abbreviations used. Additionally, how the previously reported pathogenic mutations have been described and cross-referenced to the relevant databases and guidelines for clarity and proper attribution.

4. Figures of the sequencing chromatograms:

Thank you for this excellent suggestion. We have now included sequencing chromatogram figures of the newly identified mutations to visually support our findings. We are very grateful for your detailed and constructive feedback, which has greatly helped us improve the quality, precision, and presentation of our manuscript.

---

## [Decision Letter · Decision Letter 1]

29 Oct 2025

Complete Molecular Spectrum of β-Globin Gene Mutations via Direct Sequencing Identifies Seven Novel Variants in β-Thalassemia Major

PONE-D-25-35418R1

Dear Dr. Othman,

We’re pleased to inform you that your manuscript has been judged scientifically suitable for publication and will be formally accepted for publication once it meets all outstanding technical requirements.

Kind regards,

Nejat Mahdieh

Academic Editor

PLOS ONE

Additional Editor Comments (optional):

Reviewers' comments:

Reviewer's Responses to Questions

**Comments to the Author**

1. If the authors have adequately addressed your comments raised in a previous round of review and you feel that this manuscript is now acceptable for publication, you may indicate that here to bypass the “Comments to the Author” section, enter your conflict of interest statement in the “Confidential to Editor” section, and submit your "Accept" recommendation.

Reviewer #3: All comments have been addressed

2. Is the manuscript technically sound, and do the data support the conclusions?

Reviewer #3: Yes

3. Has the statistical analysis been performed appropriately and rigorously? 

Reviewer #3: Yes

4. Have the authors made all data underlying the findings in their manuscript fully available?

Reviewer #3: Yes

5. Is the manuscript presented in an intelligible fashion and written in standard English?

Reviewer #3: Yes

6. Review Comments to the Author

Reviewer #3: The authors have carefully and satisfactorily responded to comments. The revised version demonstrates significant improvement in both structure and content. I find the manuscript ready for publication in its present form.

7. PLOS authors have the option to publish the peer review history of their article (what does this mean? ). If published, this will include your full peer review and any attached files.

**Do you want your identity to be public for this peer review?** For information about this choice, including consent withdrawal, please see our Privacy Policy .

Reviewer #3: **Yes: ** Mahdieh Soveizi

---

## [Editor Report · Acceptance letter]

PONE-D-25-35418R1

PLOS ONE

Dear Dr. Othman,

I'm pleased to inform you that your manuscript has been deemed suitable for publication in PLOS ONE. Congratulations! Your manuscript is now being handed over to our production team.

Kind regards,

on behalf of

Dr. Nejat Mahdieh

Academic Editor

PLOS ONE